# Recombinant Reg3α Prevents Islet β-Cell Apoptosis and Promotes β-Cell Regeneration

**DOI:** 10.3390/ijms231810584

**Published:** 2022-09-13

**Authors:** Luting Yu, Liang Li, Junli Liu, Hao Sun, Xiang Li, Hanyu Xiao, Martin Omondi Alfred, Min Wang, Xuri Wu, Yan Gao, Chen Luo

**Affiliations:** 1School of Life Science and Technology, China Pharmaceutical University, Nanjing 210009, China; 2School of Pharmaceutical Sciences, Nanjing Tech University, Nanjing 210037, China; 3MeDiC Program, The Research Institute of McGill University Health Centre, Division of Endocrinology and Metabolism, Department of Medicine, McGill University, Montreal, QC H3A 0G4, Canada; 4Institute of Primate Research, End of Karen Road, Karen, Nairobi P.O. Box 24481-00502, Kenya; 5State Key Laboratory of Nature Medicines, China Pharmaceutical University, Nanjing 210009, China; 6Institute of Suzhou Biobank, Suzhou Center for Disease Prevention and Control, Suzhou 215007, China; 7Suzhou Institute of Advanced Study in Public Health, Gusu School, Nanjing Medical University, Suzhou 210029, China

**Keywords:** regenerating protein, diabetes, the islets, β-cell regeneration, GRP78

## Abstract

Progressive loss and dysfunction of islet β-cells has not yet been solved in the treatment of diabetes. Regenerating protein (Reg) has been identified as a trophic factor which is demonstrated to be associated with pancreatic tissue regeneration. We previously produced recombinant Reg3α protein (rReg3α) and proved that it protects against acute pancreatitis in mice. Whether rReg3α protects islet β-cells in diabetes has been elusive. In the present study, rReg3α stimulated MIN6 cell proliferation and resisted STZ-caused cell death. The protective effect of rReg3α was also found in mouse primary islets. In BALB/c mice, rReg3α administration largely alleviated STZ-induced diabetes by the preservation of β-cell mass. The protective mechanism could be attributed to Akt/Bcl-2/-xL activation and GRP78 upregulation. Scattered insulin-expressing cells and clusters with small size, low insulin density, and exocrine distribution were observed and considered to be neogenic. In isolated acinar cells with wheat germ agglutinin (WGA) labeling, rReg3α treatment generated insulin-producing cells through Stat3/Ngn3 signaling, but these cells were not fully functional in response to glucose stimulation. Our results demonstrated that rReg3α resists STZ-induced β-cell death and promotes β-cell regeneration. rReg3α could serve as a potential drug for β-cell maintenance in anti-diabetic treatment.

## 1. Introduction

Destruction and dysfunction of islet β-cells exist in different types of diabetes and remain a major challenge for the cure of diabetes. Maintenance of functional β-cells was likely achieved by a short-term intensive insulin therapy, which is recently reported as not providing benefit on islet β-cells in Type 2 diabetic (T2D) patients [1]. The fasting-mimic therapy was proved, restoring pancreatic β-cell mass by the mechanism of inducing β-cell neogenesis, rather than a direct protection on the preexisting β-cells [2]. Expansion of islet β-cells through self-replication also displays an extremely low rate in adult humans and rodents [3]. At present, islet transplantation is the only approach available to supplement insulin-secreting cells in patients, while insufficient number of donors and immune rejection limit its clinical application [4]. Searching for agents to prevent β-cell loss is urgent for future anti-diabetic therapy.

Regenerating gene (*Reg*) was first discovered in the regenerating pancreas from 90%-depancreatized rats [5]. In the past decade, Reg protein has been identified as a trophic factor which is associated with proliferation, survival, and regeneration in diverse cells and tissues [6,7,8]. We previously demonstrated that upregulated Reg expression in response to IGF-I deficiency results in pancreatic islet enlargement and significant resistance to diabetes [9,10]. Islet-specific overexpression of Reg3β ameliorates STZ-induced diabetes [11], and Reg2 knockout leads to glucose intolerance in aging and HFD-fed obese mice [12]. These results inspired the thought that Reg proteins could be applied for the treatment of diabetes. However, one should be concerned that the expression of Reg proteins in these transgenic animal models cannot mimic the drug delivery in the treatment. Whether exogenous administration of Reg protein has therapeutic effects against diabetes needs more investigation.

Members of Reg family proteins share similar characteristics, e.g., structural module of calcium-dependent lectin domain (C-type lectin), carbohydrate binding site, three pairs of disulfide bonds, and lack of glycosylation [6]. However, the functions of the Reg isoforms are distinctive, respectively. We have thus far prepared recombinant proteins of Reg2, -3α, and -3β, and found them with different biological features. The protective recombinant Reg2 (rReg2), which also acts as an islet autoantigen, has dimorphic effects which may deteriorate islet β-cell damage under Reg2-hyperreactive immune conditions [13]. Recombinant Reg3β (rReg3β) simply exhibits a protection on β-cells by Bcl-2 and -xL upregulation [14]. Notably, in the present study of recombinant Reg3α (rReg3α), we observed increased number of scattered insulin-producing cells and clusters in the exocrine pancreas, suggesting a unique role of rReg3α that induces β-cell regeneration.

Various cells in the pancreas have been proved as valuable candidates for β-cell regeneration [15]. Constituting approximately 90% of the total pancreatic mass, acinar cells are thought to be an excellent source for β-cell neogenesis [16]. It has been reported that re-expression of Ngn3, Pdx1, and Mafa in the pancreas leads to a conversion of differentiated acinar cells into the cells resembling β-cells and other endocrine cell types [17,18]. Combination of epidermal growth factor (EGF) and ciliary neurotrophic factor (CNTF) administration stimulates acinar-to-beta cell transdifferentiation via Stat3 signaling-mediated Ngn3 expression [19]. In Reg family proteins, human Reg3A is involved in a Reg3A-Jak2/Stat3 positive feedback loop in pancreatic cancer cells [20] and mouse Reg3α and Reg4 have crosstalk with EGFR/Akt pathway [21,22]. Therefore, it is possible that rReg3α resembles EGF and the scattered small insulin-expressing cell clusters may originate from acinar cell reprogramming.

Among the prepared recombinant Reg proteins [13,14,22], we presume that rReg3α is the most potential seed for drug development because of the unique characteristic of inducing β-cell regeneration. To figure out the cell source of the neogenesis, we used isolated acinar cells labeled with wheat germ agglutinin (WGA) for cell lineage tracing. On the other hand, β-cell dedifferentiation has currently been indicated as the predominant cause contributing to pathological dysfunction of the islets in diabetes [23]. In islet β-cells, the dedifferentiation is closely related to excessive insulin demand and massive proinsulin misfolding in the endoplasmic reticulum (ER) [24,25]. Accordingly, the molecular chaperone of GRP78 that assists protein folding in the ER should be protective. Of interest, upregulation of GRP78 expression was also detected in the islets from rReg3α-treated mice, which possibly contributed to the mechanism of β-cell protection. The present study reveals the effects of rReg3α on the islets in diabetic mice, which would provide evidence that rReg3α is a valuable candidate for future anti-diabetic drug development.

## 2. Results

### 2.1. rReg3α Stimulates MIN6 Cell Proliferation

It has been reported that Akt and Erk pathways are involved in Reg signaling stimulating cell proliferation [26,27,28,29]. Whether exogenous rReg3α treatment stimulates β-cell replication remains unclear. As shown in a dose-dependent manner, the cell viability of MIN6 cells was increased by rReg3α treatment, which was attenuated under high concentrations with 1% FBS at 48 and 72 h (Figure 1A,B). The mitogenic effect of rReg3α was further confirmed by EdU staining (Appendix A). In the treatment of 100 nM rReg3α, cell cycle assay showed that the proportion of the cells in S and G2 phases increased from 11% to 26% (Figure 1C,D) and cyclin D1 and CDK4 levels were 3 and 2.5 times higher than the control (Figure 1E,F). Phospho-ATF-2 was also detected as endogenous Reg3α is reported to increase phospho-ATF-2 which binds to *cyclin D1* promoter to activate gene expression. In agreement with early reports [26,27,28,29], rReg3α induced approximately 2-, 3.5- and 7-fold increases of phosphorylated Erk, Akt, and ATF-2 (Figure 1G,H), and antagonists to Akt and Erk efficiently abolished the proliferative effect (Figure 1I). These results indicate that rReg3α is bioactive in activating Akt and Erk phosphorylation, inducing insulinoma MIN6 cell proliferation. 

### 2.2. rReg3α Protects MIN6 Cells from STZ-Induced Cell Apoptosis

The effect of rReg3α in β-cell protection was also investigated in MIN6 cells. In a challenge with STZ, the remaining healthy cells took less than 20% of the total cell number, which was rescued to 51% by rReg3α treatment. Reduction of late apoptotic (from 44% to 11%) and necrotic cells (from 13% to 1%), while accumulation of early apoptotic cells (from 24% to 37%), were found, possibly owing to a prevention of cells entering late apoptotic stage (Figure 2A,B). In cells challenged with STZ, caspase-3 cleavage was found twofold higher than the control, but this was significantly attenuated by rReg3α treatment. A fourfold elevation of Akt phosphorylation and 10- and 2.5-fold increases of Bcl-2 and -xL contents were detected with significant differences compared to the STZ group (Figure 2C,D), which may contribute to the protective mechanism [21,22].

### 2.3. rReg3α Alleviates STZ-Induced β-Cell Loss in BALB/c Mice

To further assess the in vivo effect in protecting islet β-cells, BALB/c mice were pretreated with rReg3α and then challenged with STZ. Blood glucose and bodyweight were recorded and the differences in variation were compared using AUC. Vehicle (PBS) treatment was set as the control, which exhibited as normal. STZ induced a rapid onset of hyperglycemia that peaked at 25 mmol/L, which was significantly reduced to 15 mmol/L by the treatment of rReg3α (Figure 3A,B). The acute weight loss in the STZ group reached 16 g, whereas in the STZ+rReg3α group, it was only 7.2 g (Figure 3C,D). An obvious serum insulin deficiency was detected in the STZ group, 8.3 μIU/mL, down to a half of the normal amount, which was partially restored to 12 μIU/mL by rReg3α treatment (Figure 3E). 

In the pancreatic sections, severe β-cell diminution was observed in the STZ group, which was, however, much alleviated by rReg3α treatment (Figure 3F,G). The β-cell mass in the STZ+rReg3α group was 66 mg/kg b.w., significantly higher than that of 45 mg/kg b.w. in the STZ group (Figure 3H). By semiquantitative densitometric analysis, the total insulin densities per total pancreatic area, referred to as “relative pancreatic insulin density”, were approximately 10, 4.2, and 6.9 (1000 × IOD/mm^2^), respectively, in the control, STZ, and STZ+rReg3α groups, suggesting a preservation of insulin content in the pancreas by rReg3α treatment (Figure 3I). Islet α-cell percentage (per islet cells) increased and the cells apparently migrated towards the central part of the islets when mice were challenged with STZ. No significant difference in α-cell percentage was found between the STZ and STZ+rReg3α groups (Figure 3F,J).

### 2.4. rReg3α Induces Insulin-Producing Cell Neogenesis in the Exocrine Pancreas

After carefully examining the pancreatic sections, we found a few scattered small insulin-producing cell clusters distributed in the exocrine part from the STZ+rReg3α mice, but none in the control, and seldom in the STZ group (Figure 4A). To distinguish these cells from regular islets, we scanned the pancreatic sections in the control mice and found that the size of the smallest islet structure was 425 μm^2^ (Figure 4A, up-left), leading to a speculation that the insulin-expressing clusters with size less than 425 μm^2^ may originate from neogenesis. None of these cells was accompanied by glucagon-producing cells (Figure 4B), suggesting a very low probability that the views with scattered insulin-producing cells were the cross sections of the islet edge. The number of these small insulin-producing clusters in the STZ+rReg3α group was 24/cm^2^ (<2% of the total β-cell mass), which was significantly higher than that of 5.1/cm^2^ in the STZ group (Figure 4C). Using the 425 μm^2^ threshold, the “islets” in the STZ+rReg3α mice could be classified into two subgroups, the “small insulin-producing clusters”, with a size smaller than 425 μm^2^, and the “regular islets”, which were larger than 425 μm^2^ (Figure 4D). By excluding the putative neogenic β-cell clusters below 425 μm^2^, the data of the regular islets perfectly complied with normal distribution (Figure 4E), supporting that there were two distinct subgroups of the cells and one could come from β-cell neogenesis. Statistical analysis showed that the relative insulin density in the small insulin-producing clusters reached only a half of the regular islets, 314 vs. 524 (1000 × IOD/mm^2^) (Figure 4F), which agrees with low expression of insulin in young β-cells [30]. 

### 2.5. rReg3α Promotes Insulin-Producing Cell Neogenesis in Isolated Mouse Acinar Cells

It is reported that WGA specifically binds to N-acetyl glucosamine on the plasma membrane and can be used for acinar cell lineage tracing [31]. Immunofluorescent staining in the pancreatic section was performed to evaluate the labeling of WGA. Despite a strong background in the cytosol in all cell types, it is obvious that only the exocrine cell membrane was outlined with green fluorescence, and not the endocrine part (Appendix A). In the ex vivo experiment, WGA could not reach the cytosol because the labeling was applied in live cells with integral membrane structure. Thus, the few contaminated β-cells were WGA^-^ while WGA^+^/Insulin^+^ indicated the cells that originate from β-cell neogenesis. By the treatment of rReg3α, the proportion of insulin^+^/WGA^+^ cells were 27% and 34%, respectively, at d 3 and 5, which were significantly increased compared to the control (Figure 5A,B). As auto-reprogramming occurred with a higher frequency at d 5, we selected the time point d 3 for the following examination. The mRNA levels of *Ins1* and *Ins2* increased by 3.5- and threefold (Figure 5C) and the content of secreted insulin rose from 0.16 to 0.66 ng/μg DNA under low glucose condition (basal). However, the insulin release was neither competent to respond to high glucose stimulation nor comparable to the level in isolated islets (Figure 5D), implying that these insulin^+^ cells were still immature and far from functional. Upregulation of Ngn3 expression with nuclear localization by rReg3α treatment was detected, 22% vs. that of 0.99% in the control (Figure 5E,F), which is in accordance with the report that re-expression of Ngn3 is critical for acinar-to-beta cell transdifferentiation [19]. In addition, 3.6- and 1.7-fold elevations of Ngn3 mRNA and protein levels were further confirmed, and 1.6-fold increase of Stat3 phosphorylation was also detected (Figure 5G–I). As it is well known that Ngn3 expression is regulated by Stat3 phosphorylation and the latter is associated with Reg signaling [19,20], our results hint that rReg3α could promote β-cell neogenesis via activating Stat3/Ngn3 signaling.

### 2.6. rReg3α Induces GRP78 Expression and Prevents β-cell Apoptosis

As the newborn insulin-producing cells were likely still young and took only <2% of the total β-cells in the pancreas, the effectiveness of these cells to control the glycemia was questioned. In BABL/c mice, rReg3α treatment posterior to STZ exhibited elevated amount of small insulin-producing cell clusters in the pancreas but had no effect in the glycemic control (Appendix A). These results reveal that the neogenic β-cells were not potent enough to fight against STZ-induced diabetes, and other protective mechanism(s) may exist and contribute to the effect of rReg3α treatment.

In the mice pretreated with rReg3α, pancreatic sections early at d 2 and 7 after STZ were obtained for histological examination. At both time points, the proportion of β-cells was found in the same tendency (Figure 6A–C) as observed at d 15 (Figure 3). The TUNEL signal significantly increased in the STZ group, which was largely alleviated by rReg3α treatment (Figure 6D–F). Of note, by STZ challenge only 3.1% of the β-cells were TUNEL^+^ at d 2, possibly due to the main effect of STZ inducing β-cell necrosis [11,14]. To disclose the underlying protective mechanism(s), various antibodies were used for immunofluorescent examination. Foxo1 was stained in all β-cells without any meaningful change between groups (Appendix A). Ki67, Aldh1a3, and Ngn3 were barely stained in the endocrine islets (Appendix A–D), revealing that those effects of mitogenetic, regenerative, and de- and re-differentiation were negative. Oct4^+^ cells were not detectable, while Sox2^+^ nuclei were observed in the peripheral of a few islets without significant difference (Appendix A).

GRP78 expression was found to be upregulated in the islets by rReg3α treatment. At d 2, the percentages of GRP78^high^ cells were 2.4%, 0.86%, and 4.2% of the total islet cells, and at d 7, those were 9.9%, 11.4%, and 27%, respectively, in the rReg3α, STZ, and STZ+rReg3α groups, all of which were significantly higher than the control (Figure 6G–I). At d 7, the expression of GRP78 was dramatically increased in the STZ group, but which missed the best window for β-cell protection in the initial few days after STZ. By a scan of hundreds of the islets from all groups, only a minority of GRP78^high^ cells were found co-stained with insulin (Figure 6J), and none was glucagon^+^ nor somatostatin^+^, implying that these GRP78^high^ cells may originate from β-cells. GRP78^high^ cells appeared at d 2 after the initial rReg3α treatment (Figure 6K), at the right time of STZ injection, which is thought to contribute to the protection on islet β-cells.

### 2.7. rReg3α Protects Against STZ-Induced Cell Apoptosis in Isolated Islets

The protective effect of rReg3α was further confirmed in freshly isolated mouse islets. Large numbers of early and late apoptotic cells, taking 25% and 11%, respectively, of the total cells, were observed in the STZ group. These apoptotic cells were significantly reduced to 20% and 3.6% by the treatment of rReg3α (Figure 7A,B). The impaired insulin secretion was also largely rescued, to half of the control, under both basal and high glucose conditions (Figure 7C). BrdU incorporation showed that rReg3α could not stimulate cell proliferation in isolated islets (Figure 7D), which is different from the results in the insulinoma MIN6 cell line (Figure 1). The inconsistent results can be interpreted as that tumor cells have a high capacity of proliferation while in isolated primary islets the turnover rate is very low. Western blotting results showed that the contents of GRP78, Bcl-2, and -xL and phosphorylated Akt were upregulated, and cleaved caspase-3 was downregulated (Figure 7E,F), suggesting that GRP78 and Akt/Bcl-2/-xL could be involved in the protective mechanism.

## 3. Discussion

Reg protein is worthy of being named as “regenerating protein” for the evidence in support of a great potential in the treatment of many diseases [6,7,8]. We previously prepared rReg3α and proved its protective effect against acute pancreatitis in mice [22]. In the present study, not only was a protection on the islets observed, but there was also a unique bioactivity in promoting β-cell neogenesis. Although the neogenic β-cells did not have a leading effect on the ameliorated hyperglycemia, rReg3α could still serve as a potential drug for β-cell protection/regeneration in future treatment.

In mouse pancreas, Reg3α is constitutively expressed in the acini and ducts and inducible in the endocrine islets in NOD mice [32]. A single-cell transcriptomic analysis demonstrated that a subgroup of Reg3A-positive acinar cells with progenitor-like characteristics exists, neighboring the islets in human pancreas [33]. The fragment of Reg3A-derived pentapeptide can regulate the differentiation of human pancreatic progenitor cells into functional β-cells [34]. In this study, our results again support the opinion that Reg3α (equivalent to Reg3A in human) is associated with the excellent plasticity of pancreatic cells and may play key roles in β-cell regeneration. As Liraglutide was recently reported inducing β-cell replacement in NOD mice [35], we compared the effectiveness of recombinant Reg proteins and Liraglutide in promoting β-cell neogenesis. Neither administration of rReg2 nor Liraglutide was effective in promoting β-cell regeneration using the same protocol of rReg3α treatment (Appendix A). In another word, rReg3α is distinct from Liraglutide and other Reg isoforms and is valuable for further pharmaceutical investigation.

The main challenge for therapies of β-cell regeneration is how to generate functional β-cells. In this study, the newborn insulin-expressing cells were seemly far from functional as the insulin signal captured was extremely weak, even using a confocal microscope (Figure 5A). The possibility that the faint insulin staining resulted from insulin internalization can be excluded because elevated endogenous *Ins1* and *Ins2* mRNA levels were detected. Although *Ins1* mRNA level was significantly increased by 3.5 times (Figure 5C), it should be noted that the control for the comparison was acinar cells which theoretically do not express insulin at all. Moreover, the increased amount of basal insulin secretion was neither comparable to the isolated islets nor responsive to high glucose stimulation (Figure 5D). Thus, rReg3α seemly resembles EGF which plays a crucial role in promoting β-cell neogenesis, in which other supplemental factors are also required for β-cell maturation [19,36,37].

It is acknowledged that the terminally differentiated cells, rather than the stem cells, account for β-cell regeneration [15]. It is also reported that a group of cells in the pancreas regress to the immature state, creating a pool of progenitor cells with potential to differentiate [38]. In the pancreatic development, multipotent progenitor cells differentiate into “tip” cells, also known as unipotent acinar progenitors, and “trunk” cells, which are bipotent progenitors committed to endocrine and ductal fates [39]. In the present study, Stat3 phosphorylation and Ngn3 upregulation were detected in the cultured acinar cells (Figure 5E–I), which is consistent with the references [19,20,21,22] and in support of rReg3α functioning through Stat3/Ngn3 pathway. Together, the islet β-cell loss in the STZ-treated mice (Figure 3 and Figure 4) could lead acinar cells to dedifferentiate into the precursor cells [40,41] which further adapt endocrine fate by rReg3α via Stat3/Ngn3 signaling. Although it has been reported that acinar-to-beta cell transdifferentiation mainly occurs in centro-acinar cells [42], without genetic lineage tracing our circumstantial evidence does not firmly support the acinar-to-beta cell transdifferentiation. Even though most of the scattered insulin-producing cells are isolated from ductal structures, we still cannot exclude the possibility that duct-to-beta cell conversion occurs and then the cells migrate apart from the ducts [43]. The low frequency of auto-reprogramming that occurred in the STZ group (Figure 4A,C) is likely caused by ablation of intercellular signals from islet β-cells [40]. Whether these newborn insulin-producing cells aggregate into the islets remains unknown.

As abovementioned that the neogenic β-cells were not potent enough in maintaining the blood glucose, we tested other protective mechanisms using various antibodies in immunofluorescent examination. We found that GRP78 expression was upregulated in the islets by rReg3α treatment in vivo and in vitro (Figure 6G–K and Figure 7E,F). GRP78 is an ER molecular chaperone upregulated in the unfolded protein response (UPR), which is reported to protect against HFD-induced diabetes by the maintenance of β-cell function [44]. In islet β-cells, STZ can induce ER stress by a direct destructive effect and a secondary exhaust of β-cells owing to hyperglycemia. Elevated GRP78 production prevents protein aggregation and facilitates folding and degradation, thus relieving the ER stress. On the other hand, high insulin demand causes ER stress, subsequently activating UPR mechanism, leading to β-cell dedifferentiation [24]. The state of dedifferentiation confers the β-cells with a resistance to cell apoptosis [45,46]. Coincidently, Reg protein is also reported as a stress protein secreted and activated by trypsin cleavage. The polypeptides form insoluble complexes highly organized into fibrillar structure in ECM [47], which may be relevant to the upregulated GRP78 expression. Of note, Reg proteins are also reported to be involved in accumulation and activation of macrophage subsets [48,49]. Whether the protection on β-cells is related to immunocyte regulation deserves further study.

In summary, rReg3α administration protects islet β-cells and promotes β-cell regeneration. Upregulation of GRP78 expression and activation of Akt/Bcl-2/-xL signaling are involved in the protective mechanism. Although it is still a challenge to generate functional β-cells from terminally differentiated cells, our data suggest that rReg3α is a potential candidate for novel anti-diabetic drug development.

## 4. Materials and Methods

### 4.1. Cell Culture

Mouse insulinoma MIN6 cells were seeded at 3.0 × 10^3^/well in 96-well plates and cultured in DMEM (Wisent, Saint-Jean-Baptiste, QC, Canada) supplemented with 1% or 10% FBS (Wisent), 100 U/mL penicillin, and 100 μg/mL streptomycin (Wisent). The medium contained 4.5 g/L D-glucose in all experiments, except for starvation condition with 1% FBS and 1.0 g/L D-glucose. A total of 0–500 nM rReg3α was added and cells were harvested at 24, 48, and 72 h. Cells treated with PBS were set as the control. For cell cycle assay, MIN6 cells were seeded at 1.0 × 10^6^/well in 6-well plates and cultured with 1% FBS. Amounts of 10 and 100 nM rReg3α were added 24 h before PI staining using a cell cycle and apoptosis analysis kit (Beyotime, Shanghai, China). Cells were analyzed by flow cytometry and signals were captured by FACS Calibur (BD Bioscience, San Diego, CA, USA). Statistical analysis was performed using ModFit LT software (Verity Software House, Topsham, ME, USA). In the treatment of 100 nM rReg3α, cell lysates were collected at the indicated time points for Western blotting. Inhibitors against Akt (SC-66, Abcam, Cambridge, UK) and Erk (Nimbolide, Abcam) were added and cells were harvested at 24 and 48 h. Cell viability was determined by methyl thiazolyl tetrazolium (MTT) assay according to the manufacture’s instruction (Sangon Biotech, Shanghai, China).

In apoptotic assay, MIN6 cells were seeded at 1.0 × 10^6^/well in 6-well plates with 10% FBS, and treated with 100 nM rReg3α for 12 h followed by a challenge with 10 mM STZ. After 12 h incubation, cells were harvested and labeled with Annexin-V-FITC and PI (Vazyme, Nanjing, China) for flow cytometry assay. Signals were captured by FACS Calibur and statistical analysis was performed using FlowJo 7.6.1 software (BD Bioscience). Cell lysates were collected at the indicated time points for Western blotting.

### 4.2. Western Blotting

Cells were harvested and ruptured using RIPA lysis buffer (Beyotime) containing 10 mM NaF, 1 mM Na_3_VO_4_, 1 mM PMSF, and proteinase inhibitor cocktail (Merck Millipore, Billerica, MA, USA). Cell lysates were separated by SDS-PAGE and electro-transferred to pure PVDF membranes (Merck Millipore) at 200 mA for 1 h. The blocked membranes were incubated with primary antibodies and then HRP-conjugated secondary antibodies (Table 1). Membranes were washed and incubated with hypersensitive ECL reagent (Merck Millipore), and luminescent signals were captured by a ChemiDoc XRS^+^ System (Bio-rad, Hercules, CA, USA). Densitometric quantification was performed using AlphaEase software (Alpha Innotech, San Jose, CA, USA).

### 4.3. Animals

Male BALB/c mice of 12 weeks old (Comparative Medicine Center of Yangzhou University, Yangzhou, China) were fed with standard laboratory chow and water ad libitum. Fasted for 16 h, mice were randomly grouped and challenged with an intraperitoneal injection of 150 mg/kg STZ (Sigma Aldrich, St. Louis, MI, USA) freshly prepared in citrate buffer. A total of 200 μg/kg/day of rReg3α [22] was administrated for 5 consecutive days, starting from 2 d before STZ. Mice treated with PBS were set as the control. Blood glucose at random fed and bodyweight were recorded using an ACCU-CHEK Performa glucometer (Roche, Basel, Switzerland) and electronic balance. Mice were sacrificed under isoflurane anesthesia at d 2, 7, and 15, and serum was collected for insulin content determination using enzyme-linked immunosorbent assay (ELISA, EMD Millipore). Pancreata were isolated and fixed in 4% paraformaldehyde to prepare paraffin sections. In another group, mice with developed hyperglycemia (blood glucose ≥ 11.1 mmol/L in two constitutive measurements after STZ) were injected with rReg3α, 200 μg/kg/day every two days to the endpoint, to explore the effect of rReg3α against the preexisting β-cell destruction. The present study was approved by the China Pharmaceutical University Ethics Committee (Approval Code: 2019-12-007; Approval Date: 28 December 2019).

### 4.4. Histological Examination

Using a standard procedure [14], pancreatic sections were incubated with primary antibodies at the indicated dilution and then HRP-, Cy3-, or FITC-conjugated secondary antibodies (Table 1). DAPI was used to indicate the nuclei. Microscopic images were captured using a Zeiss microscope with 200× and 400× magnifications. The integrated optical density of insulin content was conducted based on IHC sections using Image-Pro Plus 6.0 software (Media Cybernetics, Rockville, MD, USA) as reported [13,48,49]. The α-cells percentage was quantified by counting the number of glucagon-positive cells then dividing by the total islet cell number.

### 4.5. Isolated Pancreatic Acinar Cells

The pancreatic acinar cells were isolated from 22–30 g male BALB/c mice using collagenase P (Roche) digestion, DNase I (Sangon Biotech), and soybean trypsin inhibitor (Sangon Biotech) [22]. Cells were resuspended in DMEM/F12 medium (Wisent) supplemented with 100 U/mL penicillin, 100 μg/mL streptomycin, and 10% FBS. After an overnight incubation, “contaminated” cells adhered were discarded, and suspended acinar cells were collected and treated with 10 mM alloxan (Sangon Biotech) for 10 min. For lineage tracing, cells were labeled with 5 μg/mL WGA-FITC (Thermo Fisher Scientific, Waltham, MA, USA) for 24 h and then treated with 100 nM rReg3α. At d 3 and 5, cells were fixed in 4% paraformaldehyde and permeabilized in 0.2% Triton X-100 followed by an incubation with anti-insulin or anti-Ngn3 and Cy5-conjugated secondary antibodies (Table 1). Microscopic images were captured using a Zeiss LSM 5 Pascal laser scanning confocal microscope with 630× magnification.

### 4.6. Quantitative Real-Time PCR Assay (qRT-PCR)

Total RNA of the cells was extracted using Trizol reagent (Sangon Biotech), and cDNA was generated using an RT-PCR kit (Vazyme). qRT-PCR was performed using a ChamQ Universal SYBR qRT-PCR Master Mix kit (Vazyme) and QuantStudio 3 Real-Time PCR Systems (Applied Biosystems, Waltham, MA, USA). The certified primers synthesized (Sangon Biotech) are presented in Table 2.

### 4.7. Glucose-Stimulated Insulin Secretion (GSIS)

Isolated acinar cells were seeded at 1.0 × 10^6^/well in 6-well plates and treated with 100 nM rReg3α for 3 d. The cultured cells were transferred into KRBH buffer (KRB buffer supplemented with 10 mmol/L HEPES, 3 mg/mL BSA) containing 0, 2.8, or 22.2 mM glucose, in order, for 1 h each. Supernatant insulin content was determined by ELISA detection kit (Mercodia, Uppsala, Sweden) and the isolated mouse islets, 10 islets/well in 24-well plates, were set as the positive control.

### 4.8. Isolated Pancreatic Islets

The islets of Langerhans were isolated from 22–30 g male BALB/c mice using collagenase P digestion after common bile duct–duodenum ligation [11]. Isolated islets were handpicked individually using a stereoscopic microscope and cultured in RPMI 1640 medium supplemented with 20% FBS, 100 U/mL penicillin, and 100 μg/mL streptomycin overnight. Ten islets/well were placed in 24-well plates and treated with 100 nM rReg3α for 12 h, followed by an exposure to 4 mM STZ for another 12 h. The cultured islets were harvested and dissociated into single-cell suspension by an incubation with 5 μg/mL trypsin and 2 μg/mL DNase at 37 °C for 10 min. Dissociated cells were labeled with Annexin V-FITC and PI using apoptosis detection kit and flow cytometry was carried out. For proliferation assay, isolated islets were treated with 100 nM rReg3α for 24 h and BrdU incorporation was performed using BrdU labeling (Calbiochem, Darmstadt, Germany). GSIS and Western blotting were performed as described above.

### 4.9. Data Statistics

Data were presented as mean ± standard error (s.e.). Statistical analysis was performed by Student’s *t*-test or one-way ANOVA using SigmaPlot 11.0 and GraphPad Prism 8.3.0. Significance was set at *p* < 0.05. Statistics of area under the curve (AUC) was performed by the trapezoidal method.

## Figures and Tables

**Figure 1 ijms-23-10584-f001:**
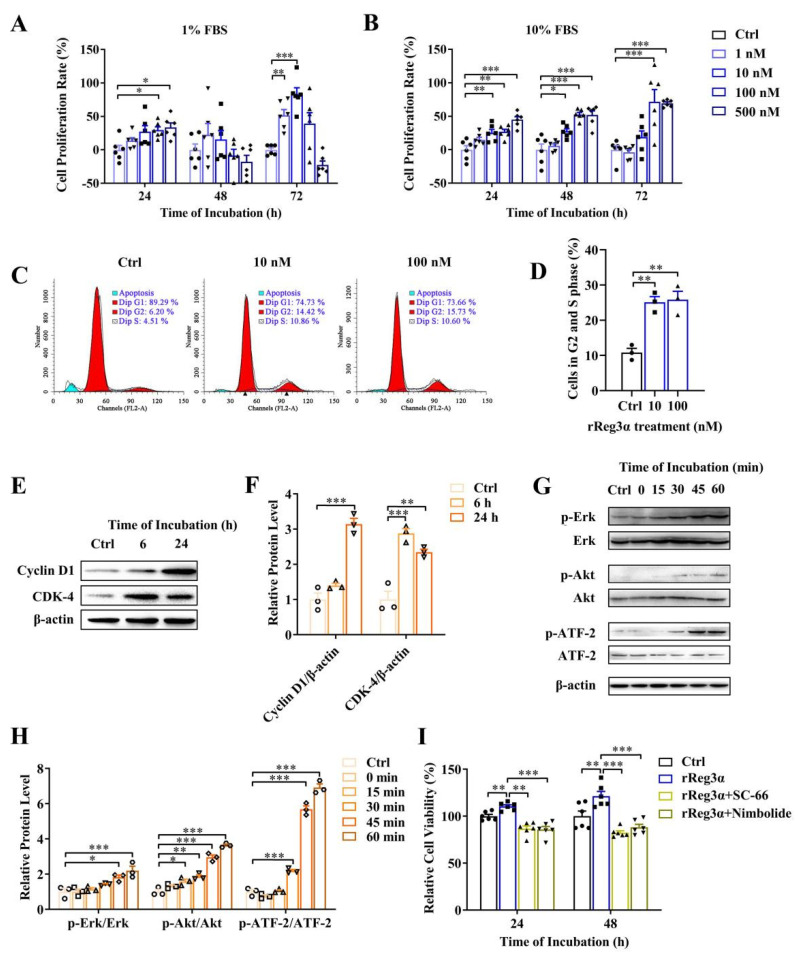
rReg3α stimulates MIN6 cell proliferation. (**A**) MTT test in MIN6 cells treated with a gradient concentration of rReg3α in 1% FBS. (**B**) MTT test in 10% FBS. * *p* < 0.05, ** *p* < 0.01, and *** *p* < 0.001 using one-way ANOVA, *N* = 6. (**C**) Cell cycle assay in MIN6 cells treated with rReg3α. Cells in S and G2 phases were considered proliferative. (**D**) Statistics of the proportion of proliferating cell in panel C. ** *p* < 0.01 using one-way ANOVA, *N* = 3. (**E**) Western blotting of CDK4 and cyclin D1 levels. (**F**) Densitometric quantification of the protein levels in panel E. The relative protein contents were corrected by β−actin. ** *p* < 0.01 and *** *p* < 0.001 using one-way ANOVA, *N* = 3. (**G**) Western blotting of Erk, Akt, and ATF−2 phosphorylation. (**H**) Densitometric quantification of the phosphorylated protein levels in panel G. The relative protein levels were corrected by the corresponding total proteins. * *p* < 0.05, ** *p* < 0.01, and *** *p* < 0.001 using one-way ANOVA, *N* = 3. (**I**) MTT test in MIN6 cells treated with rReg3α and inhibitors against Akt (SC−66) or Erk (Nimbolide). ** *p* < 0.01 and *** *p* < 0.001 using one-way ANOVA, *N* = 6.

**Figure 2 ijms-23-10584-f002:**
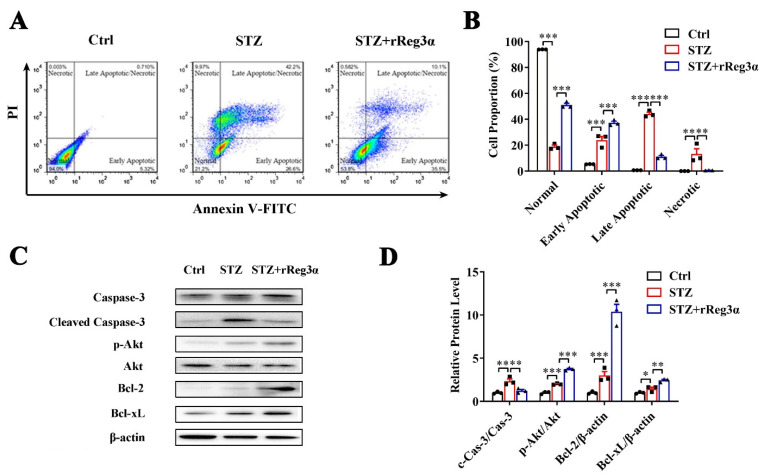
rReg3α alleviates STZ-induced MIN6 cell apoptosis. (**A**) Apoptotic assessment in MIN6 cells treated with rReg3α prior to STZ. (**B**) Statistics of the cell proportion in panel A. ** *p* < 0.01 and *** *p* < 0.001 using one-way ANOVA, *N* = 3. (**C**) Western blotting of cleaved caspase-3, phosphorylated Akt, Bcl-2, and -xL levels in MIN6 cells. (**D**) Densitometric quantification of the protein levels in panel C. The relative cleaved caspase-3 and phosphorylated Akt contents were corrected by the corresponding total proteins and the relative Bcl-2 and -xL contents were corrected by β-actin. * *p* < 0.05, ** *p* < 0.01 and *** *p* < 0.001 using one-way ANOVA, *N* = 3.

**Figure 3 ijms-23-10584-f003:**
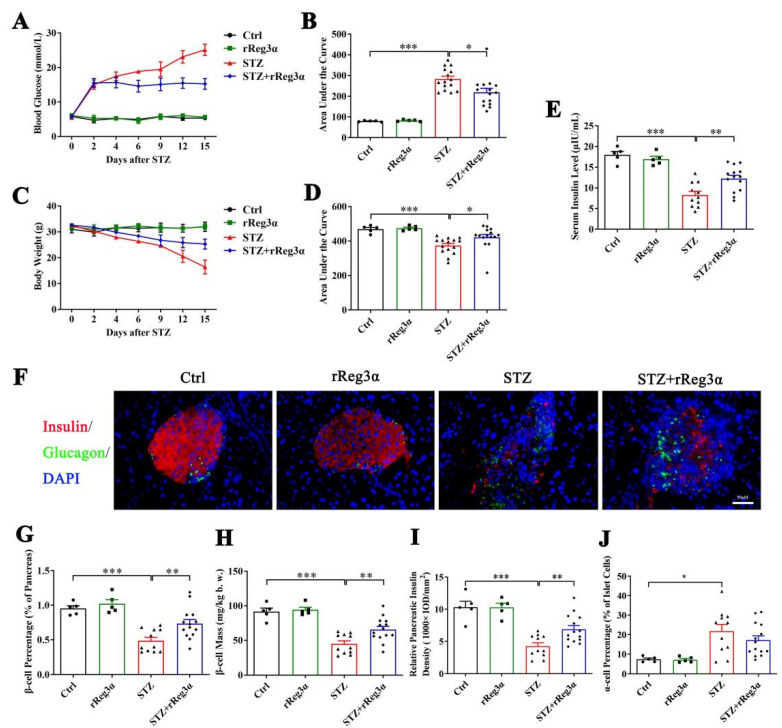
rReg3α partially rescues STZ-induced diabetes in BALB/c mice. (**A**) Change in blood glucose level within 15 d after STZ. (**B**) AUC statistics of the blood glucose in panel A. (**C**) Change in weight loss. (**D**) AUC statistics of the bodyweight in panel C. * *p* < 0.05 and *** *p* < 0.001 using one-way ANOVA, *N* = 5, 5, 15, 15. (**E**) Change of serum insulin concentration 15 d after STZ. ** *p* < 0.01 and *** *p* < 0.001 using one-way ANOVA, *N* = 5, 5, 11, 14 (a few mice died before the endpoint). (**F**) Immunofluorescent staining to insulin and glucagon (400× magnification) 15 d after STZ. Cell nuclei were labeled with DAPI. A representative image is illustrated from each group. (**G**) Statistical analysis of β-cell percentage in the pancreas. (**H**) β-cell mass per bodyweight. (**I**) Semi-quantification of insulin content in the whole pancreas. The value of “relative pancreatic insulin density” indicated the total insulin density per total pancreatic area. (**J**) α-cell percentage in the islets. * *p* < 0.05, ** *p* < 0.01, and *** *p* < 0.001 using one-way ANOVA, *N* = 5, 5, 11, 14.

**Figure 4 ijms-23-10584-f004:**
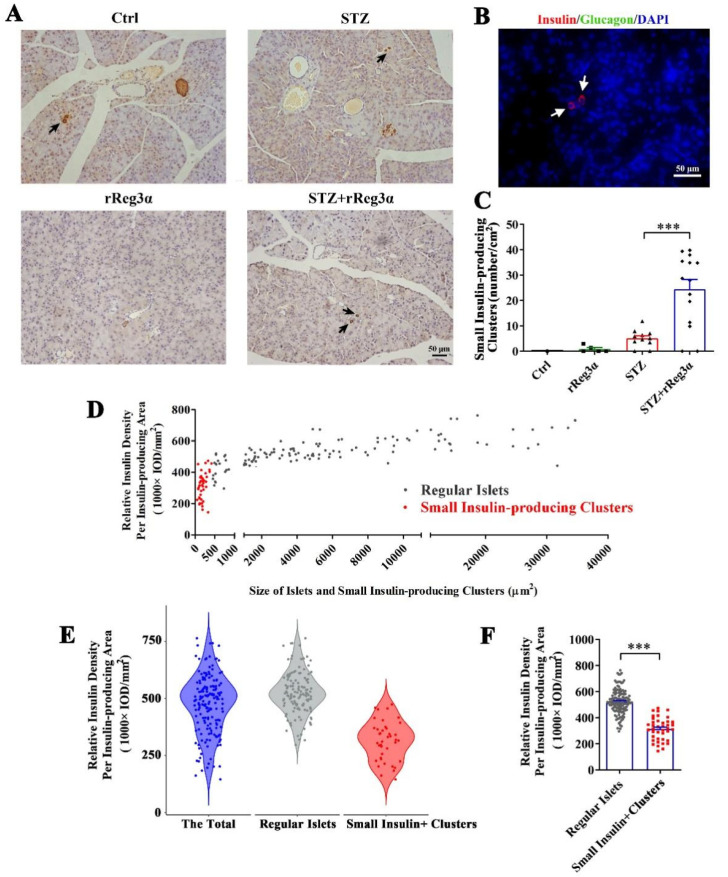
rReg3α induces insulin-producing cell neogenesis in the exocrine pancreas. (**A**) Micrographs of immunohistochemical staining to insulin (200× magnification) 15 d after STZ. A representative image is illustrated from each group. Up-left: the smallest islet, with a size of 425 μm^2^, observed in the pancreas from the control group. Insulin-producing cell clusters with the size less than 425 μm^2^ were considered to be neogenic. Black arrows indicate the putative neogenic insulin-producing cells. (**B**) Immunofluorescent staining to insulin and glucagon (400× magnification). A representative view is illustrated from all four groups. (**C**) Statistical analysis of the number of small insulin-producing clusters in panel A. *** *p* < 0.001 using one-way ANOVA, *N* = 5, 5, 11, 14. (**D**) Analysis of the regular islets (gray) and small insulin-producing clusters (red) in size and insulin density in the pancreas from STZ+rReg3α mice. (**E**) Violin plot of panel D. (**F**) Statistical analysis in panel E. *** *p* < 0.001 using one-way ANOVA. Regular Islets: insulin-stained structures with a size larger than 425 μm^2^, *N* = 133. Small Insulin+ Clusters: insulin-stained structures with a size less than 425 μm^2^, *N* = 42.

**Figure 5 ijms-23-10584-f005:**
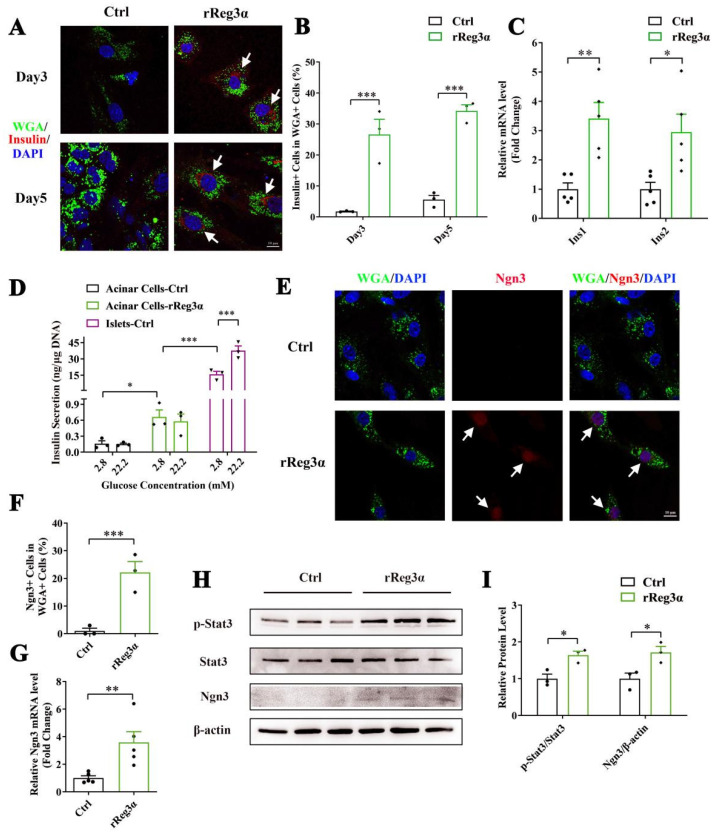
rReg3α promotes β-cell neogenesis in isolated acinar cells. (**A**) Confocal micrographs of immunofluorescent staining to insulin and WGA (630× magnification) in the isolated acinar cells treated with rReg3α at d 3 and 5. Cell nuclei were labeled with DAPI. A representative image is illustrated from each group. White arrows indicate the insulin^+^/WGA^+^ cells. (**B**) Statistical analysis of the insulin^+^/WGA^+^ cells in the total WGA^+^ cells in panel A. *** *p* < 0.001 using Student’s *t*-test, *N* = 3. (**C**) qRT-PCR analysis of *Ins1* and *Ins2* expression at d 3. * *p* < 0.05 and ** *p* < 0.01 using Student’s *t*-test, *N* = 5. (**D**) GSIS assay in the isolated acinar cells and primary islets at d 3. * *p* < 0.05 and *** *p* < 0.001 using one-way ANOVA, *N* = 3. (**E**) Confocal micrographs of immunofluorescent staining to Ngn3 and WGA (630× magnification) at d 3. Cell nuclei were labeled with DAPI. A representative image is illustrated from each group. White arrows indicate the Ngn3^+^/WGA^+^ cells. (**F**) Statistical analysis of the Ngn3^+^/WGA^+^ cells in the total WGA^+^ cells in panel E. *** *p* < 0.001 using Student’s *t*-test, *N* = 3. (**G**) qRT-PCR analysis of *Ngn3* expression at d 3. ** *p* < 0.01 using Student’s *t*-test, *N* = 5. (**H**) Western blotting of Stat3 phosphorylation and Ngn3 level in the isolated acinar cells at d 3. (**I**) Densitometric quantification of the protein levels in panel H. The relative phosphorylated Stat3 content was corrected by the total Stat3 and the relative Ngn3 content was corrected by β-actin. * *p* < 0.05 using Student’s *t*-test, *N* = 3.

**Figure 6 ijms-23-10584-f006:**
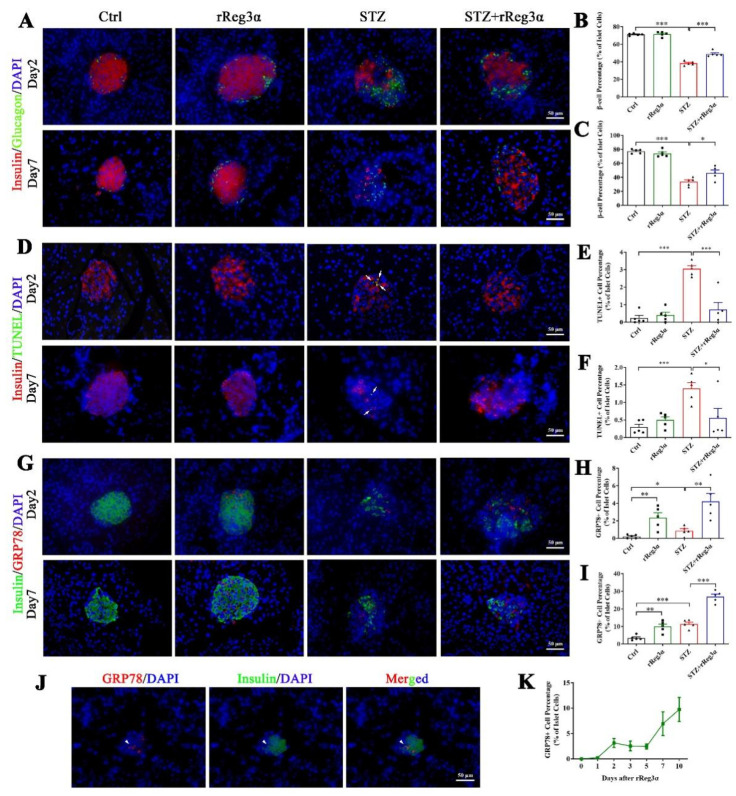
rReg3α upregulates GRP78 expression and prevents β-cell apoptosis. (**A**) Immunofluorescent staining to insulin and glucagon (400× magnification) at d 2 and 7 after STZ. Cell nuclei were labeled with DAPI. A representative image is illustrated from each group. (**B**,**C**) Statistical analysis of β-cell percentage in panel A. (**D**) Immunofluorescent staining to insulin and TUNEL (400× magnification) at d 2 and 7 after STZ. (**E**,**F**) Statistical analysis of TUNEL^+^ cell percentage in the islets in panel D. (**G**) Immunofluorescent staining to insulin and GRP78 (400× magnification) at d 2 and 7 after STZ. (**H**,**I**) Statistical analysis of GRP78^high^ cell percentage in the islets in panel G. * *p* < 0.05, ** *p* < 0.01, and *** *p* < 0.001 using one-way ANOVA, *N* = 5. (**J**) Co-staining of insulin and GRP78^high^ in a minority of endocrine cells. A representative image is illustrated from all groups. (**K**) Detection of GRP78^high^ cells within 10 d after rReg3α treatment without STZ, *N* = 3.

**Figure 7 ijms-23-10584-f007:**
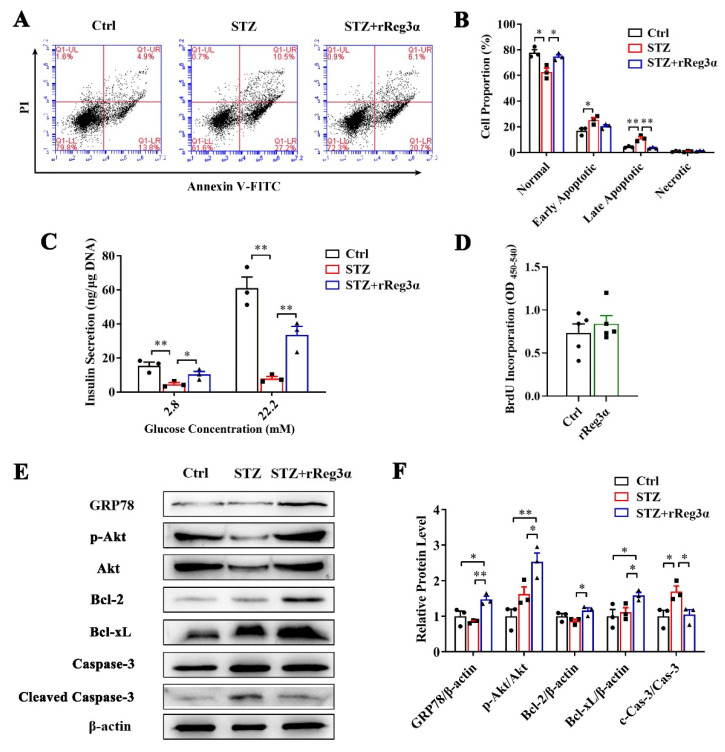
rReg3α resists STZ-induced cell apoptosis in the isolated islets. (**A**) Apoptotic assessment in the isolated islets treated with rReg3α prior to STZ. (**B**) Statistics of the cell proportion in panel A. * *p* < 0.05, ** *p* < 0.01 using one-way ANOVA, *N* = 3. (**C**) GSIS assay, * *p* < 0.05 and ** *p* < 0.01 using one-way ANOVA, *N* = 3. (**D**) BrdU incorporation, *N* = 5. (**E**) Western blotting detection of GRP78, cleaved caspase-3, phosphorylated Akt, and Bcl-2, and -xL levels. (**F**) Densitometric quantification of the protein levels in panel E. The relative GRP78, Bcl-2, and -xL contents were corrected by β-actin and the relative phosphorylated Akt and cleaved caspase-3 contents were corrected by the corresponding total proteins. * *p* < 0.05 and ** *p* < 0.01 using one-way ANOVA, *N* = 3.

**Table 1 ijms-23-10584-t001:** Antibodies used in Western blotting and histological examination.

Antibodies	Source	Identifier	Dilution (WB)	Dilution (IF)
Rabbit monoclonal to Insulin	Abcam	ab181547		1:1000
Mouse monoclonal to Insulin and Proinsulin	Abcam	ab8305		1:1250
Rabbit monoclonal to Glucagon	SAB	33385		1:500
Rabbit monoclonal to Cyclin D1	CST	2922	1:1000	
Rabbit monoclonal to CDK4	CST	12790	1:1000	
Rabbit monoclonal to phosph-Erk1/2	CST	4376	1:1000	
Rabbit monoclonal to Erk1/2	CST	4695	1:1000	
Rabbit monoclonal to phosph-ATF-2	CST	27934	1:1000	
Rabbit monoclonal to ATF-2	CST	35031	1:1000	
Rabbit polyclonal to Ngn3	Abcam	ab216885	1:1000	1:100
Rabbit monoclonal to phosph-Stat3	Abcam	ab76315	1:3000	
Rabbit monoclonal to Stat3	Abcam	ab68153	1:2000	
Rabbit polyclonal to GRP78	HuaBio	ER1706-50	1:1000	1:100
Rabbit monoclonal to Foxo1	Invitrogen	MA5-14846		1:50
Rabbit polyclonal to Ki67	Abcam	ab15580		1:200
Rabbit polyclonal to Oct4	Wanleibio	WL03686		1:50
Rabbit polyclonal to Sox2	Wanleibio	WL03767		1:50
Mouse monoclonal to Aldh1a1/2/3	Santa Cruz	sc-166362		1:200
Rabbit polyclonal phosph-Akt	Wanleibio	WLP001a	1:500	
Rabbit polyclonal to Akt	Wanleibio	WL0003b	1:1000	
Rabbit monoclonal to Bcl-2	SAB	48675	1:1000	
Rabbit monoclonal to Bcl-xL	SAB	48688	1:1000	
Rabbit monoclonal to cleaved Caspase-3	CST	9664	1:1000	
Rabbit monoclonal to Caspase-3	CST	9662	1:1000	
Rabbit monoclonal to β-actin	ABclonal	AC026	1:5000	
Goat Anti-Rabbit IgG (H&L) FITC conjugated	Zenbio	550037		1:200
Goat Anti-Mouse IgG (H&L) Cy3 conjugated	Abcam	ab97035		1:400
Goat Anti-Mouse IgG (H&L) Cy5 conjugated	Abcam	ab6563		1:1000
Goat anti-Mouse IgG (H&L) HRP conjugated	Zenbio	511103	1:7000	1:200
Goat anti-Rabbit IgG (H&L) HRP conjugated	Zenbio	511203	1:7000	1:200

**Table 2 ijms-23-10584-t002:** Primers used in mRNA expression analysis.

Target	Forward (5′–3′)	Reverse (5′–3′)
*Ins1*	CCACCCAGGCTTTTGTCAAA	CCCAGCTCCAGTTGTTCCAC
*Ins2*	GAAGTGGAGGACCCACAAGTG	CTGAAGGTCCCCGGGGCT
*Ngn3*	TGACCCTATCCACTGCTGCTT	CCTCATCCACCCTTTGGAGTT

## Data Availability

Not applicable.

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
