# Peer review of "Recombinant Reg3α Prevents Islet β-Cell Apoptosis and Promotes β-Cell Regeneration"

_ijms, 2022, doi:10.3390/ijms231810584_

Round 1
Reviewer 1 Report
Yu et al. report interesting findings on the effects of rReg3α. The manuscript is well written and presented in appropriate form. However, before being further considered a few minor and major comments need to be addressed.
Minor comments:
1. Abstract- line 24- please add "mouse" to "primary islets"
2. Abstract- line 32- what do the authors mean by the "precursor drug", unclear
3. Introduction- lines 39-40- the sentence: "(...) which was however (...)"
requires correction
4. Introduction- line 45- "insufficient donors" should be changed to "insufficient number of donors"
5. Introduction- line 63- what does it mean "seemly distinctive"?, should be "two isoforms are distinctive, respectively"
6. Introduction- lines 65-66- what does it mean "under specific conditions"- please be specific
7. Introduction- line 94- please replace "will reveal" to "reveals"
8. Introduction- line 95- "for the future anti-diabetic drug investigation" should be shorter and simplified
Major points:
1. Please correct the quality of the fluorescent/confocal microscopy pictures- particularly Figs 4-6.
2. Please comment on the increased ATF2 expression under rReg3α treatment in MIN6 cells- more information about this protein function in beta cells is required
3. How would the authors comment on the lack of mitogenic effect of rReg3α in the in vitro experiments with primary mouse islets? This contradicts observations from the STZ experiment where increased beta cell mass was noted. Please discuss. Did the authors try an alternative method to BrdU, e.g., Ki67 or PHH3 immunostainings?
Author Response
Minor comments:
Q1. Abstract- line 24- please add "mouse" to "primary islets".
[Response] We added in Abstract, page 1, line 25, “The protective effect of rReg3α was also found in mouse primary islets.”
Q2. Abstract- line 32- what do the authors mean by the "precursor drug", unclear.
[Response] We changed the confusing expression, please see page 1, line 35, “rReg3α could serve as a potential drug for β-cell maintenance in the anti-diabetic treatment.”
Q3. Introduction- lines 39-40- the sentence: "(...) which was however (...)" requires correction.
[Response] We have corrected the grammatical mistake. Please see page 1, line 43, “…, which is recently reported not providing benefit on islet β-cells in type 2 diabetic (T2D) patients.”
Q4. Introduction- line 45- "insufficient donors" should be changed to "insufficient number of donors".
[Response] Please see page 2, line 50, “…, while insufficient number of donors and immune rejection limit its clinical application.”
Q5. Introduction- line 63- what does it mean "seemly distinctive"?, should be "two isoforms are distinctive, respectively" .
[Response] We have modified the expression, please see page 2, line 67, “However, the functions of the Reg isoforms are distinctive, respectively.”
Q6. Introduction- lines 65-66- what does it mean "under specific conditions"- please be specific.
[Response] The “certain conditions” has been specified, in page 2, line 70, “which may deteriorate islet β-cell damage under Reg2-hyperreactive immune conditions.”
Q7. Introduction- line 94- please replace "will reveal" to "reveals".
[Response] Please see page 2, line 99, “The present study reveals the effects of rReg3α on the islets in diabetic mice, …”
Q8. Introduction- line 95- "for the future anti-diabetic drug investigation" should be shorter and simplified.
[Response] We have refined this sentence, please see page 3, line 101, “…, which would provide evidence that rReg3α is a valuable candidate for the future anti-diabetic drug development.”
Major points:
Q1. Please correct the quality of the fluorescent/confocal microscopy pictures- particularly Figs 4-6.
[Response] The pictures of the immunofluorescent and IHC staining were original images which lack modification. As required, we improved the quality of the images in Fig. 4 - 6. Please see the updated version in the revised manuscript.
Q2. Please comment on the increased ATF2 expression under rReg3α treatment in MIN6 cells- more information about this protein function in beta cells is required.
[Response] It is reported that endogenous Reg3α expression increases the content of phospho-ATF-2 that binds to the -57 to -52 region of cyclin D1 gene. Elevated cyclin D1 expression then promotes MIN6 cells to enter the cell cycle stimulating cell proliferation (Takasawa, Ikeda et al., 2006). Consistently, our results reveal that rReg3α treatment upregulates ATF-2 phosphorylation in MIN6 cells, suggesting the recombinant protein is bioactive and has similar biological function to the endogenous Reg3α protein.
We added the comment in Results part, please see page 3, line 113, “Phospho-ATF-2 was also detected as endogenous Reg3α is reported to increase phospho-ATF-2 which binds to cyclin D1 promoter to activate gene expression.”
Q3. How would the authors comment on the lack of mitogenic effect of rReg3α in the in vitro experiments with primary mouse islets? This contradicts observations from the STZ experiment where increased beta cell mass was noted. Please discuss. Did the authors try an alternative method to BrdU, e.g., Ki67 or PHH3 immunostainings?
[Response] (1) The MIN6 cells is a tumor cell line which derives from mouse insulinoma cells partially retaining β-cell characteristics and proliferating at a high rate. Unlike tumor cells, isolated primary islet cells gather in cluster and have an extremely low proliferative capacity. This could interpret the contradiction that rReg3α stimulates MIN6 cell proliferation but not mouse primary islets. We added the comment in Results section, page 11, line 322, “The inconsistent results can be interpreted as tumorous cells have a high capacity of proliferation but that is negative in isolated primary islets.”
(2) In the STZ experiment, the “increased” β-cell mass was mainly contributed by the protective effect against β-cell loss but not proliferation. As shown in the pancreatic sections in Suppl Fig. 5B, Ki67 staining was barely observed, revealing an extremely low turnover rate in all pancreatic cell types. Thus, in the STZ experiment we avoid saying “increased β-cell mass” but express as, for example, “Reduction of late apoptotic and necrotic cells” page 4, line 136.
(3) Although immunostaining to Ki67 and PHH3 exhibit proliferating cells more visually, there could be other problems for detection. As the intact islet is a compact sphere, the accessibility of the dyer to its core and the signal transmission could be the problems. To be honest, we currently don’t have the double-photon microscope which is required to scan the micro-organs of the islets.
Reviewer 2 Report
This paper in interesting but before publishing, it requires substantial changes either by providing strong evidence of what the authors claim and significantly change the text to avoid misinterpretation and overstatement.
1- Regarding neogenesis based on the presence of small islets, the author must provide more evidence in this regard. There is no genetic lineage tracing data included in the study. The authors neglected the presence of small or even single cells in STZ-treated mice (can be found in many studies) which are resulted from the loss of islet integrity and architecture due to cell death. My personal view on these data is that in STZ mice treated with Reg3a there is kind of protective effects and therefore the small islet clusters are bigger that those STZ mice that were not treated by Reg3a. Occurring neogenesis is still very controversial in the field and without strong evidence one needs to be very cautious to claim it.
2- Using WGA is not a proper and well accepted tool for linage tracing studies. In the filed of islet neogenesis and transdifferentiation genetic lineage tracing is more accepted and even with such methods controversial data have been reported. Claiming neogenesis and reprograming in this study required a more proper and accepted lineage tracing data. Without this, the author might consider editing the text and avoid overstatement.
3- Regarding the less function and insulin-producing of the cells that the authors claimed derived from neogenesis; STZ-treated beta cells undergo dedifferentiation process (Sachs et al 2020). How the author distinguishes the dedifferentiated beta cells compare to what they claim that come from neogenesis?
4- What exactly the authors mean when they claim the presence of new islets within exocrine or acinar. Exocrine pancreas in a single layer of epithelium. Are the new cells located within the epithelium plane or they are surrounded by mesenchymal tissue? High quality imaging with proper markers is required to show this.
5- The authors conclude that treatment Reg3 induces acinar-to-beta cell reprogramming in isolated acinar cells. To what is extend is this reproducible in vivo?
6- As a person working in this filed, I do not know any antibody that works well to detect mouse endogenous Ngn3 using western blot. Can the author provide positive control showing that such antibody works as shown in figure 5H. The same for staining in figure 5E. I don’t see proper Ngn3 signal in this figure. Better quality picture with single channel are required to support this data.
7- GRP78 is a well-characterized molecular chaperone that is ubiquitously expressed in mammalian cells. What the authors mean of % of GRP78+ cells. Does this mean there are GRP78 negative cells or only the levels of this protein changes? This should be revisited carefully. It is very hard to see any clear information on GRP78 based on figure 6G.
8- Figure 1G and subsequent quantification in H: The authors should provide a better blot of pAKT. Based on the given data, it is not possible to give quantitative statement.
9- Figure 1, the proliferation data will be better supported with some immunostaining.
10- Showing the parentage; there is no space between numbers and %. E.g. 20% not 20 %. This should be corrected through the whole manuscript.
11- line 157: "semi-quantitative densitometric analysis"
The definition of this method is not clear. Why did the authors choose to proceed in this way? Are the quantifications made based on histochemical sections or immunofluorescence?
12- The quantification of a-cells in Fig. 1J is not clear. How were cells quantified? The authors should provide a more detailed explanation of this approach. Since beta-cells are heterogeneously residing within islets FACS-analysis would be the preferred way to quantify cell-ratios.
13- Overall the authors should revise the grammar and syntax of the manuscript. The authors argumentation and conclusions are difficult to follow and comprehend in some cases and urgently need linguistic improvements.
Author Response
Q1. Regarding neogenesis based on the presence of small islets, the author must provide more evidence in this regard. There is no genetic lineage tracing data included in the study. The authors neglected the presence of small or even single cells in STZ-treated mice (can be found in many studies) which are resulted from the loss of islet integrity and architecture due to cell death. My personal view on these data is that in STZ mice treated with Reg3a there is kind of protective effects and therefore the small islet clusters are bigger that those STZ mice that were not treated by Reg3a. Occurring neogenesis is still very controversial in the field and without strong evidence one needs to be very cautious to claim it.
[Response] (1) We had the same concern that the protective effect might increase the number and size of impaired islets, which leads to a high frequency to detect the scattered insulin-producing cells. However, STZ specifically targets islet β-cells but not α-cells. If the putative neogenic insulin-producing cells are the residual β-cells from the islets losing integrity and architecture, glucagon-expressing α-cells should also be observed. As indicated in Results, page 6, line 193, “None of these cells was accompanied with glucagon-producing cells (Fig. 4B), suggesting a very low probability that the views with scattered insulin-producing cells were the cross sections of islet edge.”
Moreover, in our previous studies of rReg3β and rReg2, the protective effect on islet β-cells was observed but not the scattered insulin-producing cell clusters (Luo, Yu et al., 2016, Yu, Li et al., 2019). These results serve as the negative control indicating those cells are not the residual β-cells from destructive islets. Otherwise, we could also find scattered β-cells in rReg3β- and rReg2-treated groups. Taken together, the small insulin-producing cell clusters are different from regular islets.
(2) To the regard “there is no genetic lineage tracing data included in the study”, we have to admit that without convincing evidence the “acinar-to-beta cell reprogramming” is overinterpreted. We have modified our conclusion and the whole manuscript to avoid the misinterpretation and overstatement. For example, in Title, “Recombinant Reg3α prevents islet β-cell apoptosis and promotes β-cell regeneration”.
Q2. Using WGA is not a proper and well accepted tool for linage tracing studies. In the filed of islet neogenesis and transdifferentiation genetic lineage tracing is more accepted and even with such methods controversial data have been reported. Claiming neogenesis and reprograming in this study required a more proper and accepted lineage tracing data. Without this, the author might consider editing the text and avoid overstatement.
[Response] Indeed, genetic lineage tracing is more convincing to prove the hypothesis of acinar-to-beta cell conversion. We tried to apply for fundings to support the expense of transgenic lineage tracing model but failed. Without financial support, we alternatively used WGA staining for the in vitro experiment.
Based on the results, we still believe that the scattered insulin-producing cells originate from β-cell neogenesis. As described in Results, page 8, line 227, “It is reported that WGA specifically binds to N-acetyl glucosamine on the plasma membrane and can be used for acinar cell lineage tracing [32]. Immunofluorescent staining in the pancreatic section was performed to evaluate the labeling of WGA. Despite of a strong background in the cytosol in all cell types, it is obvious that only exocrine cell membrane was outlined with green fluorescence but not endocrine part (Suppl Fig. 2). In the ex vivo experiment, WGA could not reach the cytosol because the labeling was applied in live cells with integral membrane structure. Thus, the few contaminated β-cells were WGA- while WGA+/Insulin+ indicated the cells that originate from β-cell neogenesis.”
As acknowledged that acinar cells take up to 90% of the total pancreatic mass, it is not difficult to extract high-quality purified acinar cells. In the present study, the method for the isolation is a standard protocol which is proved efficacy to extract acinar cells with a high purity (Gout, Pommier et al., 2013). More steps were performed to ensure the accuracy of WGA-labeled cell lineage. In particular, the total extracted cells after isolation were cultured overnight to remove the adherent endothelial cells. The residual β-cells were subsequently eliminated by an incubation with alloxan. Please see Methods, page 16, line 485, “After an overnight incubation, “contaminated” cells adhered were discarded and suspended acinar cells were collected and treated with 10 mM alloxan (Sangon Biotech, Shanghai, China) for 10 min.” Moreover, the ductal cells were WGA+ but spindle-like which is distinguishable from the acinar cells. Under microscope views, we did not see much ductal cells after the adhesion and purification steps, suggesting most ductal cells were removed.
However, even though the purity of isolated acinar cells is high enough, we still only have circumstantial evidence in support of β-cell neogenesis. Regarding the suggestion from the reviewer, we should be cautious about the expression of “acinar-to-beta cell reprogramming” which we have replaced in the whole manuscript. For example, in page 8, line 251,“our results hinted that rReg3α could promote β-cell neogenesis via activating Stat3/Ngn3 signaling.”
Q3. Regarding the less function and insulin-producing of the cells that the authors claimed derived from neogenesis; STZ-treated beta cells undergo dedifferentiation process (Sachs et al 2020). How the author distinguishes the dedifferentiated beta cells compare to what they claim that come from neogenesis?
[Response] STZ was not used in the experiment of isolated acinar cells. Even if there were a few dedifferentiated β-cells, they could not be labeled with WGA. Please see the response to Q2, “Immunofluorescent staining in the pancreatic section was performed to evaluate the labeling of WGA. Despite of a strong background in the cytosol in all cell types, it is obvious that only exocrine cell membrane was outlined with green fluorescence but not endocrine part (Suppl Fig. 2). In the ex vivo experiment, WGA could not reach the cytosol because the labeling was applied in live cells with integral membrane structure. Thus, the few contaminated β-cells were WGA- while WGA+/Insulin+ indicated the cells that originate from β-cell neogenesis.”
Q4. What exactly the authors mean when they claim the presence of new islets within exocrine or acinar. Exocrine pancreas in a single layer of epithelium. Are the new cells located within the epithelium plane or they are surrounded by mesenchymal tissue? High quality imaging with proper markers is required to show this.
[Response] Indeed, without images in high resolution, we cannot say the scattered insulin-producing cell clusters are within the acini. Even though these cells are apart from duct structures, we still cannot exclude the possibility that these cells come from duct-to-beta cell conversion (Ben-Othman, Vieira et al., 2017).
We added in Discussion, page 13, line 381, “Although it has been reported that acinar-to-beta cell transdifferentiation mainly occurs in centro-acinar cells [43], without genetic lineage tracing our circumstantial evidence does not firmly support the acinar-to-beta cell transdifferentiation. Even though most of the scattered insulin-producing cells are isolated from ductal structures, we still cannot exclude the possibility that duct-to-beta cell conversion occurs and then the cells migrate apart from the ducts [44].”
Q5. The authors conclude that treatment Reg3 induces acinar-to-beta cell reprogramming in isolated acinar cells. To what is extend is this reproducible in vivo?
[Response] Without transgenic cell lineage tracing mice, we cannot provide convincing evidence in support of this conclusion in vivo. We admit that the in vivo “acinar-to-beta cell reprogramming” is overinterpreted. We have corrected this misinterpretation as mentioned in the response to Q 1-(2).
Q6. As a person working in this filed, I do not know any antibody that works well to detect mouse endogenous Ngn3 using western blot. Can the author provide positive control showing that such antibody works as shown in figure 5H. The same for staining in figure 5E. I don’t see proper Ngn3 signal in this figure. Better quality pictures with single channel are required to support this data.
[Response] We also struggled with Ngn3 antibody for a long time. We cannot provide a positive control except the instruction of this commercial antibody, please see https://www.abcam.cn/neurogenin3ngn-3-antibody-ab216885.html#lb. In Fig. 5H, we think that the bands in rReg3α-treated group are faint but still visible. In Fig. 5E, the quality of the images has been improved by enhancing the contrast.
Q7. GRP78 is a well-characterized molecular chaperone that is ubiquitously expressed in mammalian cells. What the authors mean of % of GRP78+ cells. Does this mean there are GRP78 negative cells or only the levels of this protein changes? This should be revisited carefully. It is very hard to see any clear information on GRP78 based on figure 6G.
[Response] Indeed, GRP78 is an ER molecular chaperone constitutively expressed in mammalian cells and upregulated under stress conditions. The description “percentage of GRP78+ cells” has been replaced by “percentage of GRP78high cells” and the quality of the images in Fig. 6G has been improved.
Q8. Figure 1G and subsequent quantification in H: The authors should provide a better blot of pAKT. Based on the given data, it is not possible to give quantitative statement.
[Response] The blot of pAkt in Fig. 1G has been updated.
Q9. Figure 1, the proliferation data will be better supported with some immunostaining.
[Response] As required, we carried out EdU assay to enhance the proliferation data. The result is presented in Suppl. Fig. 1. Please see page 3, line 109, “The mitogenic effect of rReg3α was further confirmed by EdU staining (Suppl Fig. 1).”
Q10. Showing the parentage; there is no space between numbers and %. E.g. 20% not 20 %. This should be corrected through the whole manuscript.
[Response] The writing format of % has been corrected and the whole manuscript has been carefully revised.
Q11. line 157: "semi-quantitative densitometric analysis" The definition of this method is not clear. Why did the authors choose to proceed in this way? Are the quantifications made based on histochemical sections or immunofluorescence?
[Response] This method is a semi-quantification of the integrated optical density of insulin content in the pancreas using Image-Pro Plus 6.0 software as reported (Bindom, Hans et al., 2010, Huang, Farmer et al., 2011, Yu et al., 2019). The quantification is based on IHC sections as the signal is more stable than fluorescence.
We added more details of this method and cited the references in Methods, page 16, line 476, “The integrated optical density of insulin content was conducted based on IHC sections using Image-Pro Plus 6.0 software (Media Cybernetics, USA) as reported [26, 49, 50].”
Q12. The quantification of a-cells in Fig. 1J is not clear. How were cells quantified? The authors should provide a more detailed explanation of this approach. Since beta-cells are heterogeneously residing within islets FACS-analysis would be the preferred way to quantify cell-ratios.
[Response] The details of the quantification have been supplemented in Methods, page 16, line 478, “The α-cells percentage is quantified by counting the number of glucagon-positive cells then divided by the total islet cell number.”
It is true that FACS analysis is preferred for cell component determination within the islets. However, other information such as the location of α-cells could be neglected. Moreover, the endocrine cells take only 1% of the total pancreas mass and we need isolate the primary islets before FACS. In the isolation the exocrine cells are discarded, which could lead to a problem to calculate the β-cell mass/pancreas.
Q13. Overall the authors should revise the grammar and syntax of the manuscript. The authors argumentation and conclusions are difficult to follow and comprehend in some cases and urgently need linguistic improvements.
[Response] The manuscript has been carefully revised to make the points clear. The conclusion has been rewritten and the changes are highlighted by “Track Changes” function of MS Word.
Round 2
Reviewer 2 Report
The authors have provided some new evidence and also modified the text accordingly. Therefore, in my opinion the manuscript is suitable for publication.